# miR-1233-3p Inhibits Angiopoietin-1-Induced Endothelial Cell Survival, Migration, and Differentiation

**DOI:** 10.3390/cells14020075

**Published:** 2025-01-08

**Authors:** Veronica Sanchez, Sharon Harel, Anas Khalid Sa’ub, Dominique Mayaki, Sabah N. A. Hussain

**Affiliations:** 1Meakins-Christie Laboratories, Department of Medicine, McGill University, Montreal, QC H4A 3J1, Canadaanas.saub@mail.mcgill.ca (A.K.S.); dominique.mayaki@muhc.mcgill.ca (D.M.); 2Translational Research in Respiratory Diseases Program, Research Institute of the McGill University Health Centre, 1001 Décarie Blvd., Montreal, QC H4A 3J1, Canada; 3Department of Critical Care, McGill University Health Centre, Montreal, QC H4A 3J1, Canada

**Keywords:** angiogenesis, endothelial cells, angiopoietins, tie-2 receptors, signaling, microRNAs, cell migration, cell survival

## Abstract

Angiopoietin-1 (Ang-1) and its receptor Tie-2 promote vascular integrity and angiogenesis. MicroRNAs (miRNAs) are involved in the regulation of many cellular functions, including endothelial cell (EC) survival, proliferation, and differentiation. Several reports indicate that these effects of miRNAs on EC functions are mediated through the modulation of angiogenesis factor signaling including that of vascular endothelial growth factor (VEGF). To date, very little is known about the roles played by miRNAs in the signaling and angiogenesis promoted by the Ang-1–Tie-2 receptor axis. Our high-throughput screening of miRNAs regulated by Ang-1 exposure in human umbilical vein endothelial cells (HUVECs) has identified miR-1233-3p as a mature miRNA whose cellular levels are significantly downregulated in response to Ang-1 exposure. The expression of miR-1233-3p in these cells is also downregulated by other angiogenesis factors including VEGF, fibroblast growth factor 2 (FGF-2), transforming growth factor β (TGFβ), and angiopoietin-2 (Ang-2). The overexpression of miR-1233-3p in HUVECs using specific mimics significantly attenuated cell survival, migration, and capillary-like tube formation, and promoted apoptosis. Moreover, miR-1233-3p overexpression resulted in reversal of the anti-apoptotic, pro-migration, and pro-differentiation effects of Ang-1. Biotinylated miRNA pull-down assays showed that p53 and DNA damage-regulated 1 (PDRG1) is a direct target of miR-1233-3p in HUVECs. The exposure of HUVECs to Ang-1, angiopoietin-2 (Ang-2), fibroblast growth factor 2 (FGF2), vascular endothelial growth factor (VEGF), or transforming growth factor β (TGFβ) triggers the regulation of PDRG1 expression. This study highlights that miR-1233-3p exerts inhibitory effects on Ang-1-induced survival, migration, and the differentiation of cultured ECs.

## 1. Introduction

Vascular development is a complex multi-step process controlled by a number of secreted vascular growth factors, such as the family of angiopoietins [1]. The angiopoietins, of which angiopoietin-1 (Ang-1) and angiopoietin-2 (Ang-2) are the best characterized, signal through the receptor tyrosine kinase, Tie-2. Ang-1 is the primary stimulatory ligand of Tie-2 receptors which upon ligation triggers the autophosphorylation of these receptors and the recruitment and activation of the phosphatidylinositol-3-kinase (PI3K) and the mitogen-activated protein kinase (MAPK) pathways [2,3]. During embryonic development, Ang-1 promotes vascular development. Indeed, Ang1^−/−^ mice exhibit embryonic lethality after E9.5 and display decreased branching, increased dilation, reduced vascular complexity, and loss of heart trabeculation [4,5,6]. In adult vasculature, Ang-1 enhances endothelial cell (EC) integrity and survival [7] and is required for the correct organization and maturation of new vessels. It also promotes quiescence and structural integrity through inhibition of apoptosis and vascular leakage [8].

Endothelial cell (EC) dysfunction contributes to various pathologies, including atherosclerosis, hypertension, and diabetes. Cardiovascular diseases remain a leading cause of morbidity and mortality despite significant advances in treatment [9].

Previously dismissed as “transcriptional noise,” non-coding RNAs (ncRNAs) now represent approximately 98% of the mammalian transcriptome and are recognized as crucial players in disease development [10,11]. Among them, microRNAs (miRNAs)—a small class of endogenous ncRNAs—have emerged as key regulators of fundamental vascular processes, including angiogenesis. miRNAs are transcribed from the genome as a primary transcript (pri-miRNA), which is processed by a micro-processing complex into a ~70-nucleotide hairpin structure known as the precursor miRNA (pre-miRNA) [12,13]. The pre-miRNA is then exported to the cytoplasm, where it undergoes further cleavage near the hairpin loop, producing a ~22-base-pair double-stranded molecule called the mature miRNA. This mature miRNA is loaded into one of the Argonaute proteins to form the RNA-induced silencing complex (RISC) [14,15]. Within the RISC, the miRNA duplex is unwound into two single-stranded molecules: the guide strand and the passenger strand. The passenger strand is typically degraded, while the guide strand binds to target mRNA, inhibiting its translation and regulating gene expression.

Numerous studies have highlighted significant alterations in miRNA expression profiles associated with various vascular diseases [16]. There is also evidence that angiogenesis factors, such as vascular endothelial growth factor (VEGF), regulate specific networks of miRNAs that play major roles in the angiogenesis responses elicited by these factors [17]. Gain- and loss-of-function studies have revealed crucial roles for specific miRNAs in regulating angiogenic processes such as endothelial cell (EC) survival, extracellular matrix production, and responses to hypoxia [18,19,20]. However, the roles of miRNAs in Angiopoietin-1 (Ang-1)-induced angiogenesis remain largely unexplored.

We hypothesized that the Ang-1–Tie-2 signaling axis promotes angiogenesis by upregulating a specific set of pro-angiogenic miRNAs while downregulating anti-angiogenic miRNAs. In this study, we investigated the regulation of miR-1233-3p by Ang-1 and its role in Ang-1-induced angiogenic responses in cultured ECs. Previously [21], we conducted high-throughput screening of miRNA expression in human umbilical vein endothelial cells (HUVECs) exposed to Ang-1 for 12 to 48 h. Our findings showed that Ang-1 exposure significantly downregulated several miRNAs, including miR-1233-3p. Building on these results, the current study confirms the downregulation of miR-1233-3p by Ang-1 and demonstrates that the overexpression of miR-1233-3p inhibits Ang-1-induced endothelial cell survival, migration, and capillary-like tube formation.

## 2. Materials and Methods

All experiments were performed at least in triplicate.

### 2.1. Materials

Recombinant human Ang-2 and Ang-1 were purchased from R&D Systems (Minneapolis, MN, USA). Vascular endothelial growth factor (VEGF), fibroblast growth factor 2 (FGF-2), and transforming growth factor beta (TGFβ) were purchased from Bioshop (Burlington, ON, Canada). Antibodies for PDRG1 (cat# SAB4503242), β-ACTIN, and β-TUBULIN were purchased from Sigma-Aldrich (St. Louis, MO, USA). HUVECs were purchased from Lonza (Basel, Switzerland). The PDRG1 expression vector was purchased from Origene (MD, USA. RC203661). Control and miR-1233-3p mimics (mirVana^®^ miRNA mimics 4464066) and inhibitors (mirVana^®^ miRNA inhibitors 4464084) were purchased from Ambion (Austin, TX, USA).

### 2.2. Cell Culture

HUVECs were maintained in MCDB131 medium (Life Technologies, Rockville, MD, USA) supplemented with fetal bovine serum (FBS, 20%), 2 mM glutamine, endothelial cell growth supplement, gentamicin sulfate, and heparin (Invitrogen, Carlsbad, CA, USA). This medium was designated as complete medium. Cells were incubated at 37 °C and 5% CO_2_.

### 2.3. Growth Factor Treatment

HUVECs were seeded (30,000 cells per cm^2^) in complete medium. The medium was then changed to MCDB131 medium without supplements, FBS or antibiotics (basic medium). After 6 h, 2%FBS and aliquots of phosphate-buffered saline (PBS, control condition), Ang-1 (300 ng/mL), Ang-2 (300 ng/mL), FGF-2 (10 ng/mL), VEGF (40 ng/mL), or TGF-β (2 ng/mL) were added to the culture medium and the cells were collected at different time points.

### 2.4. miRNA Abundance Studies

HUVECs were lysed and tRNA extracted using Qiagen miRNA extraction kit. Reverse transcription (RT) was performed using an AB TaqMan^®^ miRNA RT kit specific to miR-1233-3p and miR-126-5p. Droplet digital PCR was performed using QuantaStudio3D with 1 µL of cDNA and analyzed using QuantStudio™ 3D AnalysisSuite™ Software (https://www.thermofisher.cn/cn/en/home/life-science/pcr/digital-pcr/quantstudio-3d-digital-pcr-system/quantstudio-3d-software.html, accessed on 17 November 2024).

### 2.5. miRNA Extraction and Quantitative Real-Time PCR

The cells were lysed with Quiazol^®^ to extract total RNA. A miRNeasy Mini Kit was used according to the manufacturer’s protocols to purify miRNAs. RT was performed using the NCode™ miRNA amplification system (Invitrogen, Carlsbad, CA, USA). Real-time PCR was performed with specific primers (Table 1), Platinum SYBR^®^ (Invitrogen), and a 7500 Real-Time PCR System. The detection of Pri-miR-1233 was performed with TaqMan^®^ assays (Applied Biosystem, Foster, CA, USA). All experiments were performed at least in triplicate. miRNA expression was determined by the C_T_ method, where C_T_ values of individual miRNA data were normalized to C_T_ values of U6 snRNA as previously described [22]. miRNA-specific amplification was confirmed by a single peak in the melting curve analysis.

### 2.6. mRNA Extraction and Quantitative Real-Time PCR

Total RNA was extracted using the Ambion PureLink RNA mini kit according to the manufacturer’s protocol. Gene expression was detected using specific primers (Table 1), Power SYBR ^®^ (Invitrogen), and a 7500 Real-Time PCR System (Applied Biosystems, Foster, CA, USA). β-ACTIN was used as a housekeeping gene. All experiments were performed at least in triplicate. Relative mRNA expression was determined using the comparative threshold (C_T_) method as previously described [22]. For miRNA pull-down assays, GAPDH was used as a housekeeping gene.

### 2.7. Tie-2 Blocking Assays

HUVECs were seeded (30,000 cells per cm^2^) and were pre-treated for 1 h with 50 μg/mL of neutralizing Tie-2 IgG or control IgG antibodies. The cells were then incubated in basic culture medium supplemented with 2% FBS and aliquots of PBS or Ang-1 (300 ng/mL). The cells were collected 24 h later.

### 2.8. Exosomes Extraction

HUVECs were seeded (30,000 cells per cm^2^) and were maintained in basic medium without FBS for 6 h. The cells were then maintained for 24 h in basic medium containing aliquots of PBS or Ang-1 (300 ng/mL). The medium was then collected and used to extract exosomes according to the manufacturer’s instructions (miRCURY™ Exosome Isolation Kit, Exiqon, Vedbaek, Denmark). miRNAs from isolated exosomes were then extracted as described above.

### 2.9. Transfection miRNA Mimics and Inhibitors

HUVECs were transfected at a confluence of 50-70% using 25 nM of control mimic, miR-1233-3p mimic (mirVana^®^, 4464066), 50 nM of control inhibitor or miR-1233-3p inhibitor (mirVana^®^, 4464084). Mimics and inhibitors were transfected using Lipofectamine™ RNAiMAX (Invitrogen) according to the manufacturers’ instructions. Experiments were performed 48 h post transfection.

### 2.10. Cell Counting

HUVECs transfected with miRNA mimics or inhibitors (30,000 cells per cm^2^) were maintained in complete medium, basic medium containing Ang-1 (300 ng/mL), or PBS (control). The cells were counted 24 h later using a hemocytometer.

### 2.11. Caspase-3 Activity

HUVECs transfected with miRNA mimics or inhibitors were plated in 12-well plates and maintained in complete, basic medium containing Ang-1 (300 ng/mL), or PBS (control). The EnzChek^®^ Caspase-3 Assay Kit with Z-DEVD-AMC as a substrate (Molecular Probes, Eugene, OR, USA) was used to measure caspase-3 activity 24 h later.

### 2.12. Cell Migration

Cell migration was evaluated using a scratch (wound) healing assay. In brief, a 200 μL pipette tip was used to wound HUVEC monolayers. The cells were then maintained for 8 h in basic MCDB131 medium containing 2% FBS and aliquots of Ang-1 (300 ng/mL) or PBS (control). An Olympus inverted microscope and Image-Pro Plus™ software (Version 7.1, Media Cybernetics, Bethesda, MD, USA) were used to visualize and quantify the wounded areas using the following:(1)$$\%\;\mathrm{wound}\;\mathrm{healing}\;=[1-\left(\;\frac{{\mathrm{wound}\;\mathrm{area}\;\mathrm{at}\;t}_{8}\;h}{{\mathrm{wound}\;\mathrm{area}\;\mathrm{at}\;t}_{0}}\;\right)\times 100$$
where $t_{0}$ is the time immediately following wounding.

### 2.13. Capillary-like Tube Formation

HUVECs transfected with miRNA mimics or inhibitors (12,500 cells per well) were seeded onto 96-well plates pre-coated with growth factor-reduced Matrigel^®^ and maintained for 24 h in Lonza EGM™-2 medium (without VEGF) containing aliquots of Ang-1 (300 ng/mL) or PBS (control). The wells were imaged with an Olympus inverted microscope (40× magnification). Angiogenic tube formation was determined by measuring total tube length and average tube length using the Angiogenesis Analyzer macro for Fiji/ImageJ [23].

### 2.14. Proliferation

Transfected HUVECs (7,000 cells per well) were maintained in complete Lonza EGM™ medium in the absence of VEGF. One hour post plating, we performed a bromodeoxyuridine (BrdU) assay (Millipore, Burlington, MA, USA). The absorbance of BrdU was measured 24 h later.

### 2.15. Immunoblotting

HUVECs were lysed with RIPA buffer and total cell lysate protein levels were then measured. Immunoblotting was performed by separating denatured total cell lysates using SDS-polyacrylamide gel electrophoresis (PAGE). Proteins were then transferred onto polyvinylidene difluoride membranes. Membranes were blocked with 5% (*w/v*) low-fat milk and were then incubated with primary antibodies overnight at 4 °C. After washing, the membranes were exposed to horseradish–peroxidase–conjugated secondary antibodies followed by specific protein detection using enhanced chemiluminescence reagents (Pierce, CA). Band intensities were quantified using ImageJ software (Version 1.54m).

### 2.16. Biotin-Labeled Pull-Down Assays

The biotin-labeled miR-1233-3p mimic was designed according to Wang et al., and Cloonan et al., and ordered from IDT (Skokie, IL, USA) [24,25]. Pull-down with target mRNAs was performed as described earlier with some modifications [25,26]. Briefly, HUVECs were transfected with 50 nM of biotinylated control mimic or biotinylated miR-1233-3p mimic. After 48 h, a hypotonic lysis buffer (100 mM KCl, 5 mM MgCl_2_, 20 mM Tris-Cl pH 7.5, 5 mM DTT, 0.3% NP-40, 60 U/mL RNase OUT) containing 1× Complete Mini protease inhibitor (Roche, Basel, Switzerland) was used to lyse the cells. Magnetic streptavidin beads (Dynabeads M-280 Streptavidin, Invitrogen) coated with bovine serum albumin (1 μg/μL) and yeast tRNA (1 μg/μL) were then incubated with cell lysate for 2 h at 4 °C using a rotating platform. Cell debris was then cleared by centrifugation for 15 min at 10,000× *g* and cleared lysates were then incubated with the pre-coated beads rotating overnight at 4 °C. The beads were then washed, and RNA was released by using TRIzol and RNase-free water. RNA was then precipitated with the chloroform–isopropanol method. Reverse transcription and qPCR were then used for detection of various transcripts.

### 2.17. Statistical Analysis

Statistical analyses were performed using Graph Pad Prism 5.0 software (GraphPad® Software Inc., San Diego, CA, USA). In experiments where more than two groups were compared, two-way ANOVA followed by Bonferroni post hoc analysis was utilized. For experiments where only two groups were compared, a paired Student’s T-test was employed. Differences were considered statistically significant at *p* < 0.05.

## 3. Results

### 3.1. Regulation of miR-1233-3p by Ang-1

Figure 1 illustrates the time course of miR-1233-3p expression in HUVECs in response to Ang-1 exposure. Ang-1 increased miR-1233-3p expression after 4 h of exposure; however, the expression of this miRNA decreased significantly after 6, 24, 48, and 72 h of exposure to Ang-1 (Figure 1A,B). Unlike the effect on mature miR-1233-3p levels, Ang-1 exposure had no effect on the expression of pri- and pre-miR-1233 levels (Appendix A). Interestingly, the expression of miR-1233-5p, the sister arm of miR-1233-3p, was also significantly decreased in response to Ang-1 exposure (Appendix A). These data suggest the Ang-1 regulates the expression of both mature forms of miR-1233; however, Ang-1 had no effect on the levels of the precursor or the primary transcripts of miR-1233. mirR-1233 is an intronic miRNA located within GOLGA8A and GOLGA8B, two highly similar genes from the same family. To further test the possibility of Ang-1-regulated miR-1233-3p through the transcriptional activation of GOLGA8A, the expression of the GOLGA8A mRNA was measured after stimulation with Ang-1 (300 ng/mL). Ang-1 exposure had no effect on GOLGA8A mRNA levels as compared to PBS (Figure 1C).

Exosomes contain a variety of molecules including miRNAs and exosomal miRNA levels have been reported to be significantly altered under various physiological conditions [27,28,29]. To evaluate whether the downregulation of mature miR-1233-3p levels in response to Ang-1 exposure was due to its increased secretion in exosomes, expression was measured in exosomes isolated from HUVECs exposed to Ang-1 for 24, 48, and 72 h. Figure 1D shows that the levels of miR-1233-3p in the exosomes were not altered by Ang-1 exposure.

The importance of Tie-2 receptors in the inhibitory effect of Ang-1 on miR-1233-3p expression was assessed by using a selective blocking the Tie-2 antibody. Exposure to Ang-1 for 24 h significantly decreased miR-1233-3p expression in the presence of the control antibody but not in the presence of the blocking of the Tie-2 antibody, confirming that Ang-1 exposure decreases miR-1233-3p levels through Tie-2 receptor signaling (Figure 1E).

To determine the effects of other angiogenic factors on the expression of miR-1233-3p, HUVECs were exposed to Ang-2, FGF-2, VEGF, or TGFβ for 24 h. TGFβ and FGF-2 significantly increased while VEGF and Ang-2 significantly decreased miR-1233-3p expression (Figure 1F).

To evaluate the abundance of miR-1233-3p in ECs, we used droplet digital PCR. In HUVECs, the most abundantly expressed vascular miRNA is miR-126-5p with ~6500 copies/µL. By comparison, miR-1233-3p only had an average of ~85 copies/µL suggesting that this miRNA is not abundantly expressed in ECs (Appendix A). MiR-1233-3p is expressed in several other cell types including human microvascular endothelial cells (HMEC-1), airway epithelial cells baring mutation in the *CF2* gene, monocytes (THP1), B-lymphocytes (Ramos), and lung epithelial cells (Ramos) (Appendix A).

### 3.2. Regulation of EC Survival by miR-1233-3p

Ang-1 promotes survival, migration, differentiation, and inhibits apoptosis in ECs^4^. To assess the role of miR-1233-3p in Ang-1-induced angiogenic processes, HUVECs were transfected with mimics or inhibitors and exposed to Ang-1 (300 ng/mL) or PBS (control). The effects of the miR-1233-3p mimics and the inhibitors were verified after transfection by measuring the levels of HOXB3, a direct target of miR1233-3p (Appendix A). In cells transfected with control mimic, serum deprivation (basic medium+PBS) caused a significant decrease in cell numbers relative to complete medium (Figure 2A). Ang-1 triggered an increase in cell numbers relative to PBS (Figure 2A). In cells transfected with the miR-1233-3p mimic and maintained in complete medium, cell numbers were significantly lower than those observed with the control mimic (Figure 2A). The stimulatory effect of Ang-1 on cell numbers was still present in cells transfected with the miR-1233-3p mimic; however, absolute cell numbers remained significantly lower than with the control mimic (Figure 2A). We measured caspase-3 activity to investigate whether changes in apoptosis relate to the ameliorative effects of mimics on total cell numbers. In cells transfected with the control mimic, caspase-3 activity increased under serum deprivation. The addition of Ang-1 attenuated this increase in caspase-3 activity (Figure 2B). In cells transfected with the miR-1233-3p mimic, caspase-3 activity increased under serum deprivation caspase-3 activity in cells transfected with the miR-1233-3p mimic (Figure 2B).

A specific miR-1233-3p inhibitor was used to assess the regulatory effects of endogenous miR-1233-3p on EC survival. With the control inhibitor, cell numbers decreased, and caspase-3 activity increased under serum deprivation (Figure 2C,D). With the miR-1233-3p inhibitor, cell numbers in complete medium increased significantly as compared to the control inhibitor (Figure 2C). In addition, with the miR-1233-3p inhibitor, cell numbers decreased under serum deprivation, as compared to complete medium, but were higher than with the control inhibitor (Figure 2C). With the control inhibitor, caspase-3 activity increased under serum deprivation, as compared to complete medium and the presence of Ang-1 attenuated this response (Figure 2D). With the miR-1233-3p inhibitor, caspase-3 activity under serum deprivation increased as compared to complete medium and the presence of Ang-1 attenuated this increase (Figure 2D). With the miR-1233-3 inhibitor, caspase-3 activities measured with complete medium, basic medium, and basic medium+Ang-1 were significantly lower than those measured in the same conditions in cells transfected with the control inhibitor (Figure 2D). These results indicate that endogenous levels of miR-1233-3p exert inhibitory effects on EC survival and stimulatory effects on caspase-3 activity.

### 3.3. Regulation of EC Migration by miR-1233-3p

In cells transfected with the miR-1233-3p mimic and maintained in basic medium+ PBS, cell migration decreased relative to the control mimic (Figure 2E,F). Ang-1 increased cell migration in the presence of the control mimic but not in the presence of the miR-1233-3p mimic (Figure 2E,F). In cells transfected with the miR-1233-3p inhibitor, cell migration increased in the presence of PBS or Ang-1 relative to the corresponding values measured in cells transfected with the control inhibitor, indicating that endogenous miR-1233-3p inhibits cell migration (Figure 2G).

### 3.4. Effects of miR-1233-3p on EC Differentiation

In cells transfected with the control mimic, Ang-1 increased EC tube length by ~1.5-fold relative to PBS (Figure 3A,B). In the miR-1233-3p mimic-transfected cells, Ang-1 failed to increase average tube length when compared to PBS (Figure 3A,B).

To evaluate whether endogenous miR-1233-3p levels influence EC differentiation, tube formation was measured in control inhibitor- and miR-1233-3p inhibitor-transfected cells exposed to PBS (control) or Ang-1. Relative to PBS exposure, average and total tube lengths increased significantly in control mimic-transfected cells exposed to Ang-1 (Figure 3C,D). In miR-1233-3p inhibitor-transfected cells, total tube length measured in the presence of PBS increased significantly as compared to cells transfected with the control inhibitor (Figure 3C,D). In miR-1233-3p inhibitor-transfected cells, Ang-1 failed to increase average and total tube lengths as compared to PBS (Figure 3C,D).

### 3.5. Regulation of EC Proliferation by miR-1233-3p

The effects of Ang-1 on proliferation are debatable as it has been described as having a positive effect, a negative effect, and no effect [7]. Therefore, we tested the effect of miR-1233-3p on EC proliferation in the absence of Ang-1. We found that the overexpression of miR-1233-3p with mimics had no major effects on BrdU incorporation. By comparison, the inhibition of miR-1233-3p in HUVECs using a selective inhibitor triggered a significant increase in BrdU incorporation suggesting that endogenous miR-1233-3p inhibits EC proliferation (Figure 3E,F).

### 3.6. PDRG1 Is a Direct Target miR-1233-3p

miRNAs regulate gene expression by destabilizing mRNA and by inhibiting mRNA translation. To identify mRNAs targeted by miR-1233-3p in ECs, we used three algorithms, such as DIANA, TargetScan, and miRDB, which predicted that PDRG1 is a possible direct target of miR-1233-3p (Figure 4A). To confirm this prediction, HUVECs were transfected with miR-1233-3p mimics or inhibitor and the expression of PDRG1 was measured. The overexpression of miR-1233-3p significantly reduced PDRG1 mRNA expression while transfection with the miR-1233-3p inhibitor increased PDRG mRNA expression (Figure 4B). Similarly, the PDRG1 protein level significantly decreased in miR-1233-3p mimic-transfected cells and increased in miR-1233-3 inhibitor-transfected cells (Figure 4C–F). According to the DIANA database, the 3′UTR region of PDRGG1 mRNA contains two conserved miR-1233-3p binding sites (nucleotides 816-823 and 209-215) (Figure 5A). To confirm the direct binding of miR-1233-3p to PDRG1 mRNA, HUVECs were transfected with biotinylated control or miR-1233-3p mimics and then biotin-labeled pull-down assays were performed as described in the methods section. Total RNA was harvested from the pull-down and input materials, and PDRG1 mRNA levels were measured using qPCR. The efficiency of miRNA-1233p3p overexpression using the biotinylated mimic was confirmed (Figure 5B,C). We found that PDRG1 mRNA levels were significantly enriched in the pull-down material of cells transfected with biotinylated miR-1233-3p as compared to cells transfected with the control mimic (Figure 5D). The mRNA levels of centromere protein B (CENPB), another predicted target of miR-1233-3p, were also enriched in the pull-down materials whereas those of zinc finer protein 91 (ZFP01), not a predicted target of miR-1233-3p, were not enriched in the pull-down material (Figure 5D,E).

Previous studies have revealed that intronic miRNAs may regulate the expression of their own genes. For instance, it has been reported that miR-26b, which is transcribed from an intron of CTDSP2, directly targets and regulates the expression of CTDSP2 and that this regulation is important for neuronal differentiation during development [30]. In this study, we report that GOLGA8A mRNA expression is not affected by miR-1233-3p overexpression, suggesting that this miRNA does not regulate the expression of GOLGA8A (Appendix A).

### 3.7. Regulation of PDRG1 Expression by Ang-1

Our results indicate that Ang-1 exposure triggers significant downregulation of miR-1233-3p in ECs and that this miRNA directly targets PDRG1. To assess whether PDRG1 expression is regulated by the Ang-1–Tie-2 axis, we measured PDRG1 mRNA and protein levels in HUVECs exposed to Ang-1 for 2 to 72 h. Ang-1 exposure triggered a significant increase in PDRG1 mRNA after 12 and 48 h and PDRG1 protein levels after 24 h (Figure 6A–D). We also observed that other growth factors, including Ang-2, FGF-2, and VEGF, upregulated the expression of PDRG1 protein levels in HUVECs (Figure 6E,F). These results suggest that the Ang-1–Tie-2 axis downregulates miR-1233-3p expression to enhance PDRG1 expression by removing the inhibitory effect of miR-1233-3p on PDRG1 mRNA. These effects on miR-1233-3p and PDRG1 expression are not selective to the Ang-1–Tie-2 axis since similar results were observed in response to Ang-2 and VEGF exposure (Figure 1 and Figure 6).

## 4. Discussion

The main findings of this study are the following: (1) the Ang-1–Tie-2 axis decreases the expression of miR-1233-3p levels in ECs; (2) The overexpression of miR-1233-3p is associated with the significant inhibition of Ang-1-induced migration, proliferation, and the survival of ECs; (3) endogenous miR-1233-3p levels exert a negative effects on EC survival, migration, capillary-like tube formation, and proliferation; (4) PDRG1 mRNA is a direct target of miR-1233-3p; and (5) Ang-1 upregulates PDRG1 at the mRNA and protein levels.

It has been well established that pro- and anti-angiogenic stimuli regulate angiogenesis through the modulation of miRNA levels. Perhaps the most well-known pro-angiogenic stimulus is VEGF, which has been found to induce, in a time-dependent manner, the expression of several angiogenic miRNAs including: miR-191, -17-5p, -18a, and miR-20a in HUVECs [31]. Treatment with epidermal growth factor (EGF) and VEGF increased the expression of miR-296a brain ECs [32]. Moreover, VEGF-induced angiogenesis has been shown to be mediated in part through the downregulation of miR-101 expression, which leads to enhanced expression of histone methyltransferase EZH2 [33]. The role of miRNAs in Ang-1-induced angiogenesis remains largely unexplored. Recently, our group reported that miR-146-5p expression is upregulated by Ang-1 in ECs and that the transfection of miR-146b-5p mimics in HUVECs significantly attenuates toll-like receptor 4 signaling through the selective targeting of IRAK1 and TRAF6 protein expressions [34]. In the current study, we describe for the first time the importance of miR-1233-3p in the regulation of angiogenesis. We found that the expression of this poorly characterized miRNA is significantly downregulated by Ang-1 in ECs and that this response is independent of the expression of the genes where miR-1233-3p resides since GOLGA8A and GOLGA8B expressions remained unchanged in response to Ang-1 exposure. We also found that the decrease in the cellular levels of miR-1233-3p mediated by the Ang-1–Tie-2 axis is not the result of increased exosomal export. However, we do not exclude the possibility of miR-1233-3p being exported by other secretory mechanisms like microparticles or lipoproteins known to be intercellular miRNA carriers [35,36,37].

Our study indicates that the overexpression of miR-1233-3p using a mimic leads to an increase in basal caspase-3 activity and a significant decrease in EC survival, migration, and proliferation. These results in ECs are in agreement with those derived from trophoblast cells by Zhong et al., who reported that overexpressing miR-1233-3p significantly decreased the proliferation and invasion abilities of trophoblast cells [38]. These authors also described that the expression of miR-1233-3p is significantly elevated in the placental tissue of hypertensive disorder-complicated pregnancies. Another study reported that a significant correlation exists between elevated serum miR-1233-3p levels and the development of pre-eclampsia [39]. In another study, the significant elevation of miR-1233-3p levels in the tissues of patients with renal cell carcinoma (RCC) was reported [40].

Of the numerous predicted targets of miR-1233-3p, only HOXB3 has been validated using a luciferase construct containing the 3′ UTR of this mRNA [38]. HOXB3 is a member of the mammalian *HOX* genes located on one of the four homologous *HOX* loci (*HOXA, B, C, D*). Several reports suggest that HOX genes play a role in promoting or inhibiting angiogenesis [41,42,43,44,45]. For example, the constant expression of HOXD10 inhibits angiogenesis [43]. HOXB3, in particular, is upregulated during EC differentiation as documented in a human bone marrow-derived mesenchymal stem cell (hMSC) model [46]. Other studies have demonstrated that HOXB3 promotes cancer cell migration and progression when upregulated [47,48].

To identify miR-1233-3p targets, we utilized pull-down assays of a biotinylated miR-1233-3p miRNA mimic and identified direct interaction between PDRG1 mRNA and miR-1233-3 in HUVECs. Just like miR-1233-3p, PDRG1 is not a well-characterized protein. It consists of 133 amino acids with a molecular mass of 15511 Da. There is a relatively high homology between human and mouse PDRG1 protein sequences [49]. PDRG1 bears a helix–turn–helix-like motif at the C-terminal end and this motif is likely to mediate protein–protein and protein–DNA interactions. PDRG1 also has a β-prefoldin-like domain. Luo et al., showed that PDRG1 is expressed predominantly in the cytoplasm [48]. Mass spectrometry analyses have identified the R2TP/Prefoldin-like complex composed of the prefoldin and prefoldin-like proteins URI, art-27, PFD2, PFD6, and PDRG1 [50]. Together with HSP90, the R2TP/prefoldin-like complex is responsible for the assembly of the RNA polymerase II complex (pol II) in the cytoplasm of eukaryotic cells [50]. In addition, one member of this complex, URI, is involved in stabilizing PDRG1. Mass spectrometry analysis and immunoprecipitation studies have described a specific interaction between PDRG1 and URI in the nucleus in prostate cells [51]. It is also hypothesized that all the prefoldin and prefoldin-like proteins in this complex, such as URI, art-27, PFD2, PFD6, and PDRG1, interact with each other through the β strands of the prefoldin-like domain [51].

In the present study, we found that the mRNA and the protein levels of PDRG1 were elevated upon treatment with Ang-1. Other growth factors had similar effects as well. PDRG1 expressions have been shown to be elevated in several tumors including those in the colon, rectum, ovary, lung, stomach, and uterus [52]. A recent study by Sun et al. reported that PDRG1 promotes the proliferation and migration of glioblastoma multiforme both in vivo and in vitro, and that these effects are mediated by the mitogen-activated protein kinase (MEK)–extracellular regulated protein kinase (ERK)/CD44 pathway [53]. Similarly, the silencing of PDRG1 in colorectal cancer cells decreased vitality, invasion, and migration, and induced cell apoptosis and G0/G1 phase arrest [54]. PDRG1 knockdown also decreased tumor growth and metastasis and enhanced apoptosis-related protein expression [54]. While recent progress has been made in understanding the functional roles of PDRG1 in cancer cell biology, no studies to date have investigated its expression and role in regulating endothelial cell (EC) functions. Further research is needed to clarify the importance of PDRG1 in mediating the pro-angiogenic effects of Ang-1 in ECs.

## Figures and Tables

**Figure 1 cells-14-00075-f001:**
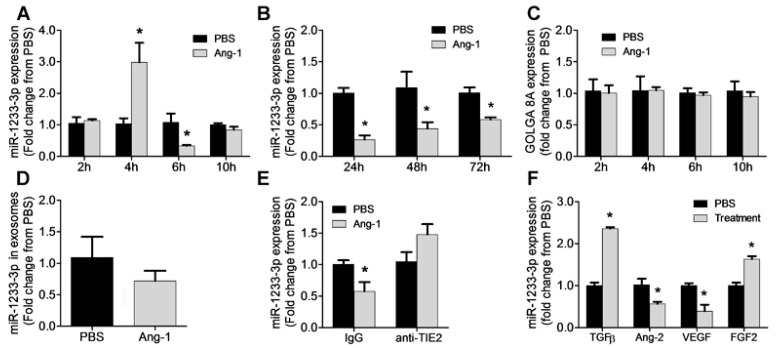
Ang-1 regulates miR-1233-3p in HUVECs. (**A**,**B**) miR-1233-3p levels measured in HUVECs exposed for 2, 4, 6, 10, 24, 48, and 72 h to PBS or Ang-1 (300 ng/mL). (**C**) Expression of GOLGA8A mRNA in HUVECs exposed for 2, 4, 6, and 10 h to PBS or Ang-1. (**D**) Levels of miR-1233-3p in HUVEC exosomes after 24 h of PBS or Ang-1 (300 ng/mL) treatment. (**E**) Expression levels of miR-1233-3p in HUVECs treated with control or Tie-2 blocking antibodies and incubated with PBS or Ang-1 (300 ng/mL) for 24 h. (**F**) Expression of miR-1233-3p in HUVECs treated PBS or growth factors (TGFβ, Ang-2, VEGF and FGF-2) for 24 h. Values are expressed as means± SEM and are expressed as fold change from PBS. * *p* < 0.05, compared with PBS.

**Figure 2 cells-14-00075-f002:**
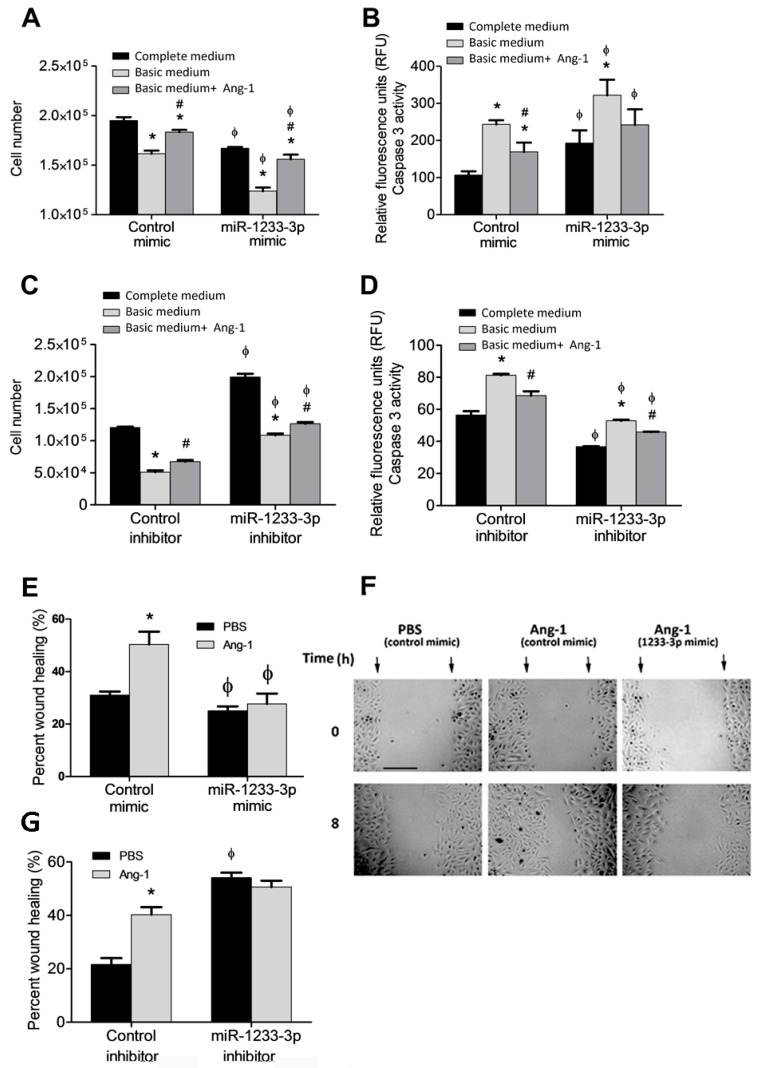
Regulation of cell survival and migration by miR-1233-3p and Ang-1. (**A**,**B**) HUVECs transfected with control or miR-1233-3p mimics were maintained in complete medium, basic medium+FBS, or basic medium + Ang-1. (**C**,**D**) HUVECs transfected with control or miR-1233-3p inhibitors were maintained in complete medium, basic medium+ FBS, or basic medium + Ang-1. (**E**,**F**) Percentage wound healing and representative images of HUVECs transfected with control or miR-1233-3p mimics and treated with PBS or Ang-1 (300 ng/mL) for 8 h. (**G**) Percentage wound healing of HUVECs transfected with control or miR-1233-3p inhibitors and treated with PBS or Ang-1 (300 ng/mL) for 8 h. Values are means ± SEM. * *p* < 0.05, compared to PBS alone. # *p* < 0.05, compared to basal medium. ϕ *p* < 0.05, compared to cells transfected with control mimic or control inhibitor. Scale bar in panel F = 100 µm.

**Figure 3 cells-14-00075-f003:**
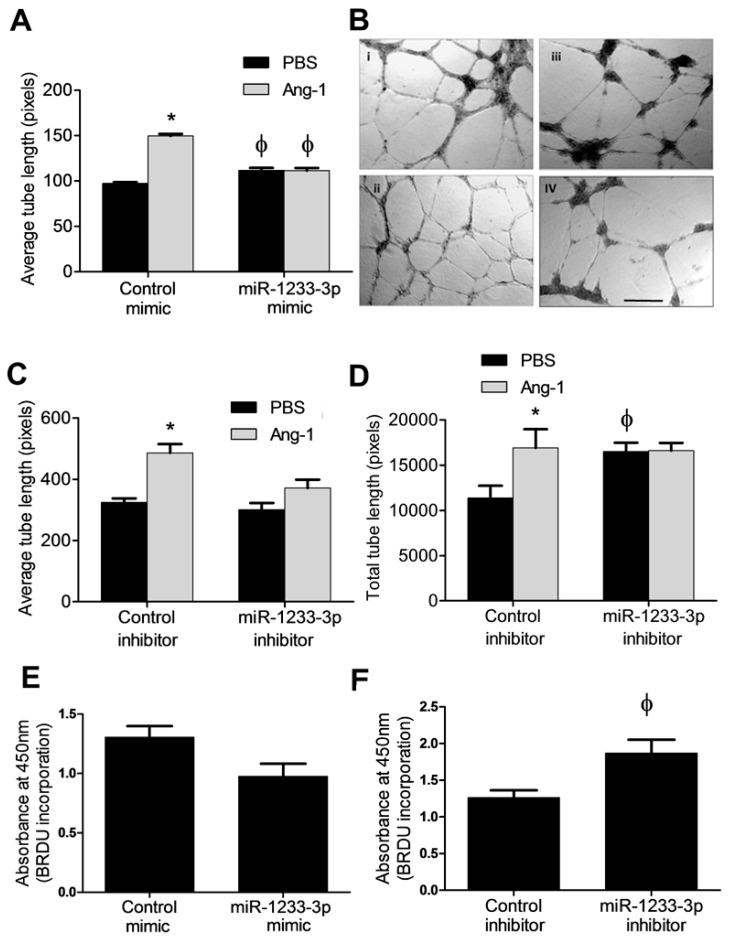
Effects of miR-1233-3p on EC differentiation and proliferation by miR-1233-3p. (**A**) The average tube lengths of capillary-like tube formation in HUVECs transfected with the control and miR-1233-3p mimic in the presence of PBS or Ang-1. Values are means ± SEM. (**B**) Representative images of capillary-like tube formation in cells transfected with control mimic in the presence of PBS (i) or Ang-1 (ii). Also shown are HUVECs transfected with the miR-1233-3p mimic in the presence of PBS (iii) or Ang-1 (iv). (**C**,**D**) The average and total tube lengths of cells transfected with the control and miR-1233-3p inhibitors in the presence of FBS or Ang-1. (**E**,**F**) BrdU incorporation in HUVECs transfected with the control and miR-1233-3p mimics or inhibitors in the presence of PBS. Values are means ± SEM. * *p* < 0.05, compared to PBS alone. ϕ *p* < 0.05, compared to cells transfected with control mimic or control inhibitor. Scale bar in panel B = 200 µm.

**Figure 4 cells-14-00075-f004:**
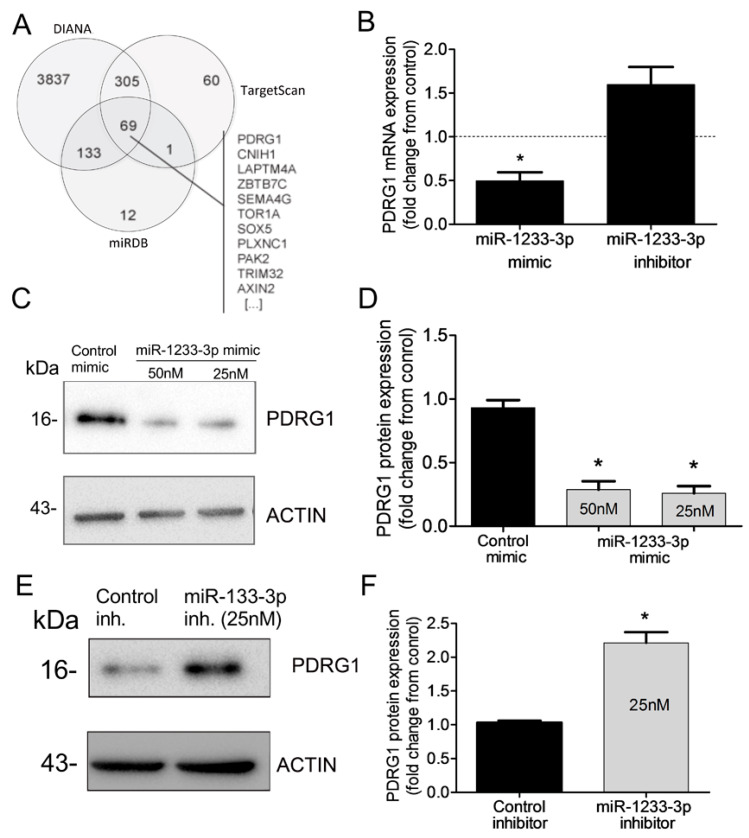
Identification of PDRG1 as miR-1233-3p target. (**A**) Venn diagram displaying in silico predicted targets of miR1233-3p as computed by DIANA, TargetScan, and miRDB algorithms. (**B**) PDRG1 mRNA expression in HUVECs transfected with miR-1233-3p mimic or inhibitor and their respective controls (denoted by dotted line). Values are expressed as fold change from control mimic and inhibitor. (**C**,**D**) Representative immunoblots of PDRG1 and β-ACTIN proteins and quantification of PDRG1 protein levels in HUVECs transfected with control or miR-1233-3p mimics. Values are means± SEM and are expressed as fold change from control mimic. (**E**,**F**) Representative immunoblots of PDRG1 and β-ACTIN proteins and quantification of PDRG1 protein levels in HUVECs transfected with control or miR-1233-3p inhibitors. Values are means± SEM and are expressed as fold change from control inhibitor. * *p* < 0.05, compared to control mimic or inhibitor.

**Figure 5 cells-14-00075-f005:**
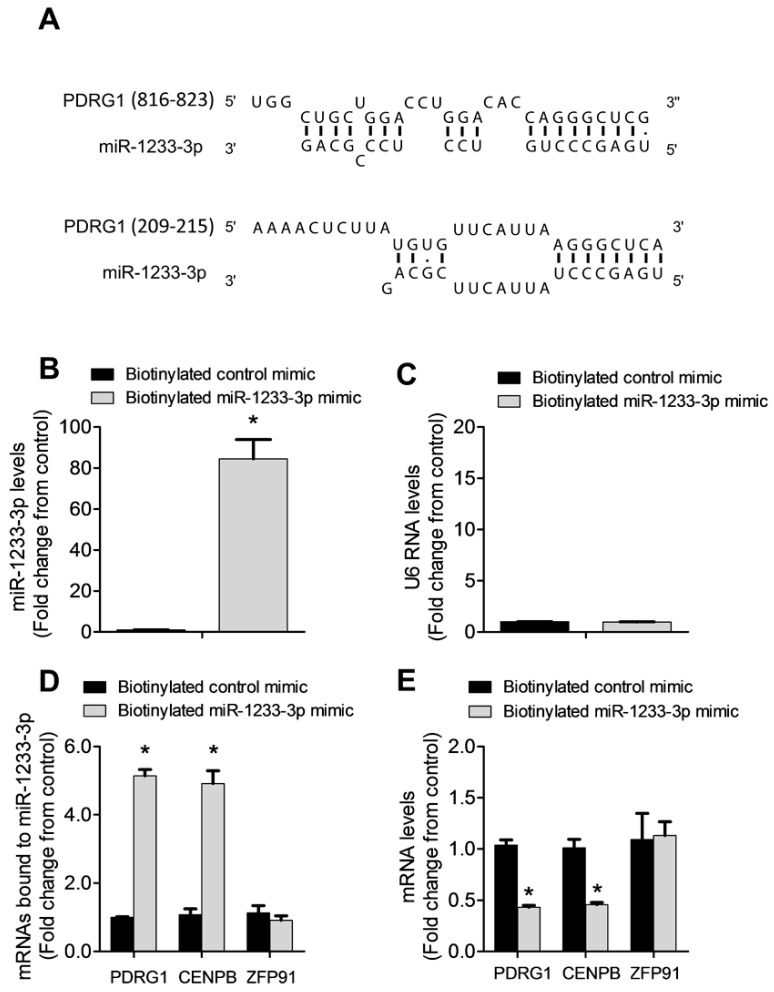
PDRG1 is a direct target of miR-1233-3p. (**A**) A schematic representation of the sequence alignment of miR-1233-3p and the predicted binding sites according to the DIANA algorithm on the 3′ UTR of PDRG1 mRNA. The numbers between the brackets indicate the nucleotide number of 3’ UTR of PDRG1. (**B**–**D**) Biotinylated miR-1233-3p pull-down assays. (**B**,**C**) Illustrate the levels of miR-1233-3p and U6 RNA after the transfection of 50 nM biotinylated control or miR mimics in HUVECs. Values are expressed as a fold change from the control mimic. (**D**) Shows mRNA levels of PDRG1, HOXB3, and ZFP91 detected with qPCR in the pull-down materials isolated from HUVECs transfected with the biotinylated control or miR-1233 mimics. Values are expressed as a fold change from the control mimic. (**E**) The levels of PDRG1, HOXB3, and ZFP91 mRNA in the total input mRNAs measured by qPCR. Enrichment was calculated as follows: X = miR pull-down/control pull-down. Y = miR input/control input. Fold binding = X/Y. Values are expressed as means ± SEM. * *p* < 0.05, compared to the biotinylated control mimic.

**Figure 6 cells-14-00075-f006:**
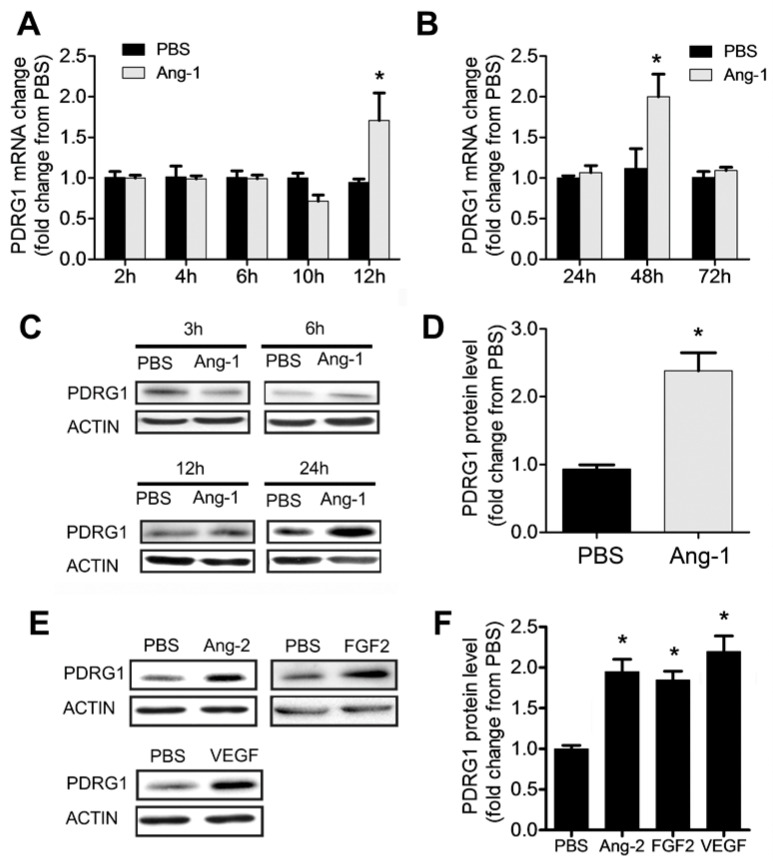
Ang-1 regulates PDRG1 expression. (**A**,**B**) PDRG1 mRNA levels in HUVECs treated with PBS or Ang-1 (300 ng/mL) for 2, 4, 6, 10, 12, 24, 48, and 72 h. (**C**) Representative immunoblots of PDRG1 protein in HUVECs exposed to PBS or Ang-1 for 3, 6, 12, and 24 h. (**D**) Mean values of PDRG1 protein levels in HUVECs exposed to PBS or Ang-1 for 24 h. (**E**,**F**) Representative immunoblots of PDRG1 and β-ACTIN proteins and mean values of PDRG1 protein levels in HUVECs exposed for 24 h to PBS, Ang-2 (300 ng/mL), FGF-2 (10 ng/mL), or VEGF (40 ng/mL). Values are expressed as means± SEM and are expressed as a fold change from PBS. * *p* < 0.05, compared to PBS.

**Table 1 cells-14-00075-t001:** Primers used for quantitative real-time PCR experiments.

Name	Sequence (5′→3′)	Accession Number	Expected Size (bp)
β-ACTIN	F: AGAAAATCTGGCACCACACC R: GGGGTGTTGAAGGTCTCAAA	NM_001101	126
GAPDH	F: AAGAAGGTGGTGAAGCAGGCG R: ACCAGGAAATGAGCTTGACAA	NM_02396.1	166
GOLGA8A GOLGA8B	F: TAGGCTCCCGTGCTTTTCT R: AAAGCTCCCCCAAAAGGTTA	NM_181077.3 NM_001023567.4	285 285
PDRG1	F: TGGGTGGTTGCTGAATGAAG R: GGTAAGAGGCGCATCCACTC	NM_030815.2	222
CENPB	F:GACGTTCCGGGAGAAGTCAC R:AGCCCTCGAGCTTGTCGTAG	NM_001810.5	215
HOXB3	F: GGTGGAGCTGGAGAAGGAGT R: GGCCTTCTGGTCCTTCTTGT	NM_002146.4	148
ZFP91	F: CTCGCTATTTGCAGCACCACR: GCCCGAGCACAATATTCACA	NM_053023.4	165
miR-1233-3p	F:TGAGCCCTGTCCTCCCGCAG	MIMAT0005588	
miR1233-5p	F:AGTGGGAGGCCAGGGCACGGCA	MIMAT0022943	
pre-miR-1233	F: AGTGGGAGGCCAGGGCACGGCA R: GCGGGAGGACAGGGCTCA	NR_036050	
U6	F: ACTAAAATTGGAACGATACAGAGA	NR_004394.1	
5S	F: ATACCGGGTGCTGTAGGCTT	D14867	

## Data Availability

All available data are included in the manuscript.

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
