# Peer review of "miR-1233-3p Inhibits Angiopoietin-1-Induced Endothelial Cell Survival, Migration, and Differentiation"

_cells, 2025, doi:10.3390/cells14020075_

Round 1

Reviewer 1 Report

Comments and Suggestions for Authors

In this study, the authors investigated the role of miRNA-1233-3p in the Ang-1/Tie2-mediated angiogenic signalling pathway using an in vitro model with HUVECs. They showed that miRNA-1233-3p targets PDRG1 and regulates apoptosis, cell migration and tube formation in endothelial cells. This study is well designed and presents very interesting results. The data presented appear to be reliable. To further improve the quality of the manuscript, the following revisions are recommended:

Comments

Please include representative microscopic images for each group in the wound healing assay and capillary-like tube formation assay in Figures 2 and 3, respectively.

Author Response

Comment 1: 

Please include representative microscopic images for each group in the wound healing assay and capillary-like tube formation assay in Figures 2 and 3, respectively.

Response 1

We agree with your suggestions. We have revised Figures 2 and 3 to include representative examples of wound healing and capillary-like tube formation. Thanks for the suggestion

Reviewer 2 Report

Comments and Suggestions for Authors

This study highlights that miR-1233-3p plays important roles in regulating Ang-1/Tie-2 signaling and angiogenesis in ECs.The concept is intriguing. Authors need to correct the manuscript throughout starting from the title. There are several minor issues (listed below) which need to be addressed. Over all, this manuscript is not recommended for publication in the current form but may be considered for publication provided the following concerns are addressed.

1. The article must point out why miR-1233-3p should be detected, and the miRNA should be obtained after miRNA Abundance Studies, and then verified.It should not be determined directly in the method section.

2. Positive controls and overexpression groups should be established.

3.The title “Role of miR-1233-3p in Angiopoietin-1-Induced Angiogenesis”involve a wide range.It is suggested that more precise title based on the research be established.The conclusion is too positive,which needs further verification and more animal experiments.

Author Response

Comment 1: The article must point out why miR-1233-3p should be detected, and the miRNA should be obtained aftermiRNA Abundance Studies, and then verified.It should not be determined directly in the method section.

Response 1: We have revised the introduction section to include a statement about our previously published study in which we detected several miRNAs whose expression was downregulated in response to Ang-1 treatment in endothelial cells. One of these miRNAs which was downregulated by Ang-1 is miR-1233. Please see the revised introduction section 

Comment 2: Positive controls and overexpression groups should be established.

Response 2: We are unclear what you meant by positive controls and overexpression groups. In the current study, we used both loss-of-function (inhibitor) and gain-of-function (mimic) of miR-1233-3p to evaluate the functional importance of this miRNA in regulating angiogenic responses elicited by Ang-1 in cultured endothelial cells

Comment 3: The title “Role of miR-1233-3p in Angiopoietin-1-Induced Angiogenesis”involve a wide range.It is suggested that more precise title based on the research be established.The conclusion is too positive,which needs further verification and more animal experiments.

Response 3: We agree with your suggestion and we have changed the title to indicate that miR-1233-3p inhibits Ang-1-induced proliferation, migration, and differentiation of in vitro cultured endothelial cells. We have also modified our conclusion in the last sentences of the abstract

Reviewer 3 Report

Comments and Suggestions for Authors

Ang-1/Tie-2 stimulat  angiogenesis by up-regulating a specific set of pro-angiogenic miRNAs and downregulating anti-angiogenic miRNAs. This work relevant to the filed, based in the results shows inhibit Ang1-induced EC survival, migration, and capillary-like tube formation by decreases the expression of miR-1233-3p levels in ECs. Exosomes extraction is novedose and transfection miRNA for show his relevance in Capillary-Like Tube Formation.

Minor comments: 
1. Is it possible that miR-1233-3p is subjected to polyadenylation?
2. What drug allows differentiation between migration and proliferation?
3. Maybe mention a little about the regulation of miRNA expression.
4. Suggest adding some more recent references.

Author Response

Comment 1:  Is it possible that miR-1233-3p is subjected to polyadenylation?

Response: All human pri-miRNAs are polyadenylated. Mature miRNAs are quite short and not polyadenylated. 

Comment 2: What drug allows differentiation between migration and proliferation?

Response 2: Many drugs can differentiate between migration and proliferation. In our study, we did not need to use any drug in this regard because we chose to measure migration 8 hours post wounding. Hence, there was no sufficient time for proliferation to contribute to wound healing and only migration is the major factor causing wound healing. There is also evidence that in human endothelial cells, migration rather than proliferation is the major factor for wound healing even if this parameter was measured 24 hours post-wounding (Ammann, DeCook and Slepian. Exp. Cell Res. 2019).

Comment 3:  Maybe mention a little about the regulation of miRNA expression.

Response 3: We agree with your suggestion and we have edited the introduction to include references about the regulation of miRNAs by angiogenesis factors including our published study in this regarding concerning Ang-1 regulation of miRNAs. 

Comment 4: Suggest adding some more recent references.

Response 4: We agree with your suggestion and we have included several new references in the reference list.